# A Highly Sensitive, Ultra-Durable, Eco-Friendly Ionic Skin for Human Motion Monitoring

**DOI:** 10.3390/polym14091902

**Published:** 2022-05-06

**Authors:** Zhaoxin Li, Haoyan Xu, Na Jia, Yifei Li, Liangkuan Zhu, Zhuangzhi Sun

**Affiliations:** 1Province Key Laboratory of Forestry Intelligent Equipment Engineering, College of Mechanical and Electrical Engineering, Northeast Forestry University, Harbin 150000, China; lizhaoxin@nefu.edu.cn (Z.L.); haoyanx@nefu.edu.cn (H.X.); jiana1975@sina.com (N.J.); yifeili@nefu.edu.cn (Y.L.); 2Key Laboratory of Bio-Based Material Science & Technology, Ministry of Education, Northeast Forestry University, Harbin 150000, China

**Keywords:** ionic skin, starch, sensitivity, motion monitoring

## Abstract

Ionic conductive hydrogels have shown great potential in areas such as wearable devices and electronic skins. Aiming at the sensitivity and biodegradability of the traditional flexible hydrogel electronic skin, this paper developed an ionic skin (S−iSkin) based on edible starch–sodium alginate (starch–SA), which can convert the external strain stimulus into a voltage signal without an external power supply. As an excellent ion conductive polymer, S−iSkin exhibited good stretchability, low hydrophilicity and outstanding electrochemical and sensing properties. Driven by sodium ions, the ion charge transfer resistance of S−iSkin is reduced by 4 times, the capacitance value is increased by 2 times and its conductivity is increased by 7 times. Additionally, S−iSkin has excellent sensitivity and linearity (R^2^ = 0.998), a long service life and good biocompatibility. Under the action of micro-stress, it can produce a voltage change ratio of 2.6 times, and its sensitivity is 52.04. The service life test showed that it can work stably for 2000 s and work more than 200 stress–voltage response cycles. These findings provide a foundation for the development of health monitoring systems and micro-stress sensing devices based on renewable biomass materials.

## 1. Introduction

Ionic skin is regarded as a most attractive candidate that has great potential to replace human skin, which can sense various external stimuli of force, temperature, illumination and humidity [1,2,3]. Conductive hydrogels (CH) including ionic types and electronic conductive types have actively promoted the development of ionic skin [4,5,6]. It is well known that ionic CH mainly includes three parts of water, polymers and ionic conductors, and its electrical behavior is profoundly affected by electrical signals through electrons and holes [7,8,9]. Therefore, the design of the conductive network of ionic CH is essential to improve its sensing performance.

At present, there is still a long way to go for ionic skin to imitate the use of real human skin. Since ionic skin is directly exposed on the surface of human skin, its biocompatibility and the environmental degradability after it is discarded must be considered. The green, degradable starch polymer has the advantages of being renewable, cheap, biodegradable and biocompatible, and provides a choice for the natural skeleton of ionic skin [10,11,12]. Considering that the types of organic solvents are still very limited, such as water, ionic liquids, polyacrylic acid and dimethyl sulfoxide [13,14,15], the high-temperature gelatinization between water and starch molecules is still a low-cost, simple process and an eco-friendly way to prepare ionic CH. Importantly, ionic CH is physically or chemically connected to construct 3D networks, which can provide basic mechanical support and ion channels [16,17]. Due to its large porosity and excellent mechanical properties, sodium alginate combined with starch can obtain a better three-dimensional skeleton support. Simultaneously, the ionic skin can realize the perception of external strains or pressure excitations, but as a real skin, it can still show high sensitivity and linearity under the action of micro-forces, which is a great challenge in the design of ion skin. At present, the strategy of the outstanding ionic conductivity is still the most direct and effective way to increase the sensitivity of ionic CH [18,19,20]. Recently, a large number of new high-conductivity polymers have been developed in polypyrrole, polythiophene, polyacetylene and polyaniline, etc., which have been widely used to improve the electrical performance of ionic CH [21,22,23,24]. In addition, the environmental durability of ionic skin needs to be considered, and the cyclical excitation response characteristics are still our key indicators for continuous enhancement. Especially in a room-temperature or high-temperature environment, the moisture in the ion CH is easily lost [25,26,27,28]. In this case, its strain sensitivity, mechanical properties and service life will be greatly affected, or even fail.

Here, we propose a self-powered sensing and ion-conducting hydrogel using highly environmentally friendly and excellent biomaterials. Based on its good adhesion and mechanical properties, it can be used for ion skin preparation strategies. In this strategy, an ion-driven ion skin (S−iSkin) based on edible starch–sodium alginate as a grid-like framework is proposed. Meanwhile, it has excellent sensitivity, linearity, a long service life and good biocompatibility. These excellent properties allow S−iSkin to be effectively applied for detection possibilities such as human health monitoring. Compared with the existing electronic skins, it can be directly attached to the human skin surface through its own adhesion without external conditions and an external power supply for motion monitoring, which makes S−iSkin work without any damage to the human body. After failure, it can become ion-conducting hydrogel again by simple high-temperature dissolution. These excellent properties indicate that our developed renewable biomass ionic skin has an important role in serving human health monitoring and in micro-stress sensing devices.

## 2. Experimental Section

### 2.1. Materials and Experiment Method

The potato starch (Starch, CAS: 9005-25-8) and polyaniline (PANI, purity ≥ 98%, CAS: 5612-44-2) were purchased from the Cool Chemical Technology Company (Beijing, China) in the experiment. Sodium alginate (SA, purity ≥ 99.9%, CAS: 9005-38-3) was ordered from Xiling Chemical Company (Shantou, Guangdong, China), and sodium chloride (NaCl, purity ≥ 99.9%, CAS: 7647-14-5) was acquired from Tianda Chemical Reagent Factory (Tianjin, China). Nano-silica (SiO_2_, particle size of about 30 nm, purity ≥ 99.5%, CAS: 7631-86-9) was purchased from Keyan Industrial Co., Ltd. (Shanghai, China), and glycerol (purity ≥ 99.9%, CAS: 56-81-5) was bought from Yongda Chemical Reagent Co., Ltd. (Tianjin, China). Deionized water (CAS: 7732-18-5) was obtained from Yongchang Reagent Company (Harbin, China).

The microstructure of S−iSkin was characterized by a cold field emission scanning electron microscope (SEM, SN-7500F, JEOL, Beijing, China) under an accelerating voltage of 5 kV. The functional groups of the test samples were analyzed using a Nicolet iS50 Fourier Transform Infrared Spectrometer (FT-IR, Thermo fisher, Waltham, MA, USA) in the spectral range of 4000–500 cm^−1^. The composition and structure of S−iSkin were carried out on an X’Pert3 Powder X-ray diffractometer (XRD, PANalytical, Almelo, The Netherlands) at a scanning range of 5–55° and a scanning speed of 5° min^−1^. The hydrophobicity test of the sample was measured by an OCA20 video optical contact angle measuring instrument (Dataphysics, German). Tensile tests were obtained by using an AG-A10T wood universal testing machine (Shimadzu, Japan) at a tensile speed of 10 mm min^−1^. The electrochemical properties of the experimental samples were established through cyclic voltammetry (CV), the alternating current impedance method (EIS) and the charge–discharge method (GCD) by using a Corrtest CS350H multi-channel chemical test station (Kesite, Zhongshan, Guangdong, China). The sensing signal of S−iSkin is obtained with Keithley 6514 electrometer electrometer (Keithley, Beaverton, OR, USA).

### 2.2. Evaluation Method of Parameters and Performance of S−iSkin

In this experiment, the electrochemical properties of S−iSkin were measured as follows. S−iSkin was analyzed by the two-electrode system according to cyclic voltammetry (CV), the alternating current impedance method (EIS) and the charge–discharge method (GCD). In the experiment, 1 mol L^−1^ LiCl solution was used as the electrolyte solution.

The area-specific capacitance of S−iSkin tested using the CV method is calculated by Equation (1).
(1)$$C=\frac{1}{2\cdot s\cdot r\cdot \Delta V}\int_{V_{0}}^{V_{0}+\Delta V}IdV$$

Whereby *s* is the surface area of the electrode, *r* is the voltage sweep rate, Δ*V* is the potential drop during the entire cycle and *V*_0_ is the lowest voltage during the cycle. *I* represents the charge and discharge current value corresponding to the potential window change.

In the EIS test, the electric double layer capacitance (*C_dl_*) and the ionic conductivity (*σ*) of S−iSkin can be obtained using Equations (2) and (3).
(2)$$C_{dl}=\frac{1}{\omega \cdot R_{ct}}$$
(3)$$\sigma =\frac{L}{R_{ct}\cdot A}$$

Herein, *ω* is the highest angular frequency of the arc, *L* is the thickness of the sample and *A* is the surface area of S−iSkin.

In the GCD test, the specific capacitance *C*, the energy density *E* and the power density *P* of S−iSkin can be obtained using Equations (4)–(6).
(4)$$C=\frac{I\cdot \Delta t}{m\cdot \Delta V}$$
(5)$$E=\frac{1}{2}\cdot C\cdot {\left(\Delta V\right)}^{2}$$
(6)$$P=\frac{E}{\Delta t}$$

Whereby *m* is the mass of the active material on the electrode, *I* is the magnitude of the charging current, Δ*V* is the charging potential difference and Δ*t* is the charging time.

In our experiment, the sensitivity of S−iSkin was analyzed to evaluate its sensing performance. The sensitivity *GF* can be obtained according to Equation (7).
(7)$$GF=\frac{\frac{\Delta V}{V}}{\frac{\Delta L}{L}}$$

Whereby *V* is the voltage value of S−iSkin in a stable condition, Δ*V* is the voltage change value of S−iSkin under the action of external stress, *L* is the initial length of S−iSkin without external stress, Δ*L* is the length change value of S−iSkin under the action of external stress. In this experiment, the length of the test sample of S−iSkin and the test stress are small, so its length change under stress is small, and the change with stress is basically the same. In this paper, Δ*V/V* is mainly used to evaluate the sensing performance of S−iSkin.

### 2.3. Fabrication Process of S−iSkin

First, two grams of potato starch were dissolved in 25 mL of deionized water under a water bath at 65 °C, and the potato starch was under constant stirring to form the green hydrogel based on the gelatinization of starch. 0.01 g silica (SiO_2_) and 1 mL glycerol were added to the hydrogel and stirred evenly, and dried in air for 24 h to form a starch film. Then, 0.5 g of sodium alginate (SA) was placed into the starch hydrogel and dried to form a starch–SA film, which effectively reduces its hydrophilic properties. On this basis, 2.0 g NaCl was added as the kinetic ion to improve its sensing performance, preparing the starch–SA film. Finally, the conductivity of the starch–SA-based ionic hydrogel is doped with 0.1 g PANI to effectively improve its conductivity, and it is dried to form a S−iSkin (for the specific preparation process, see Figure 1a).

Under the action of a single stress on S−iSkin, it bulges to the non-stressed side. This makes the free anions and cations inside S−iSkin redistribute at both of its ends according to the ion radius. Some negatively charged anions (Cl^−^) move to the side of S−iSkin away from the stress due to their large volume, while the dissociated cations (Na^+^) move to the side of S−iSkin under stress due to their small volume. Due to the movement of anions and cations and their continuous accumulation, a directional electric field is formed at both sides of S−iSkin, thereby forming a voltage change signal. As the stress on S−iSkin disappear, it gradually returns to its original state according to its strong elasticity. The anions and cations in S−iSkin are redistributed under the action of Van der Waals forces and electric field forces at this moment. Finally, they gradually return to their initial state and the voltage change signal gradually disappears (see Figure 1b).

## 3. Results and Discussion

### 3.1. Structure and Morphology of S−iSkin

The SEM image of the starch film shows the microstructure of its surface is relatively dense (see Figure 2a). The denser structure affects the conductivity of moving ions in the film. In order to improve the performance of the starch film, SA is introduced as an auxiliary component to form a starch–SA film. While this approach effectively reduces its hydrophilicity, it can also change its microstructure into a more regular network structure to facilitate the conduction process of moving ions inside (see Figure 2b). The starch–SA film is used as the skeleton, and the ionic starch–SA film is formed by introducing NaCl as the kinetic ion, which effectively improves its electrical performance (see Figure 2c). In order to further improve the electrical conductivity of the ionic starch–SA film, a S−iSkin is formed by doping PANI in it (see Figure 2d). Figure 2e,f respectively show the cross-sectional SEM images of starch film and S−iSkin under the action of a liquid nitrogen brittle fracture. It can be seen that the starch film provides a good carrier for NaCl and PANI particles, and the two are relatively loosely distributed in S−iSkin. The introduction of PANI effectively reduces the agglomeration of NaCl particles in S−iSkin, thereby improving its electrical and sensing performance.

### 3.2. Structural Characterization and Mechanical Properties of S−iSkin

The FT-IR curve analysis of S−iSkin shows that the peak distribution of the functional groups in its internal composition is similar to that of the starch film (see Figure 3a). This makes its performance closer to that of the starch film, which makes it have a better degree of degradation than the other three films. According to XRD curve analysis, the XRD curves of the starch–SA-based ionic hydrogel film and S−iSkin show obvious diffraction peaks at 31.8° and 45.5°, which confirms that there is a large amount of NaCl inside (see Figure 3b). The diffraction peak at 20.3° in S−iSkin disappears because the introduction of PANI reduces the hydrogen bond between the original molecules and enlarges the inter-molecular gap, which is beneficial to the transfer of moving ions. In addition, S−iSkin has a larger water contact angle (57.5°) compared with the original starch film (15.1°) by adding SA and PANI (see Figure 3c). This allows it to work in some humid environments, which effectively solves the limitations of the existing sensors in practical applications. The mechanical properties of S−iSkin and the other three groups of samples (length × width: 50 mm × 10 mm) are analyzed, according to the stress–strain curve shown in Figure 3d. It shows that S−iSkin has a higher tensile strength (1.35 MPa) and a better elongation at break (60%) compared with the other three groups (see Figure 3e,f). 

This is mainly because S−iSkin has a grid-like skeleton, which makes it have a large specific surface area. The introduction of PANI effectively reduced the crystallization of large NaCl particles, enabling them to be uniformly dispersed into starch–SA, thereby reducing the formation of NaCl crystal nuclei. Meanwhile, as a high-molecular polymer, the highly conductive PANI can also have a strong Van der Waals force with the high-molecular polymer (Starch and SA) [29,30], which is higher than the Van der Waals force in the starch–SA film. Therefore, under the combined action of three factors, the mechanical properties of S−iSkin have been greatly improved.

Compared with the starch–SA–NaCl film in Figure 3a, the disappearing peaks of S−iSkin at 2913 cm^−1^, 2848 cm^−1^, 1734 cm^−1^ and 1472 cm^−1^ are the *v*_C-H_ asymmetric stretching vibration peak of the unsaturated group, the *v*_C-H_ symmetric stretching vibration peak of the unsaturated group, the *v*_C=O_ stretching vibration peak of the carboxyl group and the *v*_C-C_ unsaturated ring stretching vibration peak, respectively. This is mainly due to the strong oxidizing property of PANI, which can oxidize the unsaturated functional groups in SA, so that the peaks at the above positions disappear. However, S−iSkin has a *v*_N-H_ stretching vibration peak at 1644 cm^−1^, which proves that PANI can exist relatively stably in S−iSkin. In addition, since the unsaturated functional groups of SA are largely oxidized, the performance of S−iSkin is closer to that of starch films, and its chemical properties are more stable and have a better degree of degradation.

### 3.3. Electrochemical Performance of S−iSkin

In the experiment, the CV method was used to analyze the electrochemical performance of S−iSkin, and S−iSkin was immersed into 1 mol L^−1^ of LiCl solution. The scanning voltage range was set at 0.2 V to 0.7 V, and the scanning rates were set at the range of 10 mV s^−1^–400 mV s^−1^ (for the CV curves, see Appendix A). It can be found that the CV curves under different scanning rates were rectangle and presented no obvious redox peaks. The corresponding specific capacitance per unit area of different groups at different scan rates was calculated by the area ratio of the CV curve using Equation (1). It can be seen that the specific capacitance of S−iSkin has obvious advantages both at the low scan rate and at the high scan rate (see Figure 4a). Since the specific capacitance reflected the migration efficiency of the ion movement inside the film, the electrochemical performance of S−iSkin was significantly better than the other three groups. 

Figure 4b shows the EIS curve of S−iSkin and the equivalent circuit of the test system in the scanning range of 10^5^ Hz–0.01 Hz. The results showed that the introduction of PANI and NaCl particles made S−iSkin have a lower equivalent resistance Re (2.63 Ω) and a lower charge transfer resistance Rct (3.53 Ω). So, S−iSkin obtained good ion conduction and charge transfer (see Figure 4c,d). According to Equations (2) and (3), to further analyze the Nyquist curve of S−iSkin, it can be found that the doping of PANI makes S−iSkin have a significantly excellent capacitance value of C_dl_ (0.36 mF) and an excellent conductivity of σ (0.14 mS cm^−1^). This verified that it has an excellent electrochemical performance and sensing performance compared to the other three sets of samples (see Figure 4e,f).

A galvanostatic charge–discharge (GCD) test was performed on S−iSkin and the other three sets of samples using a potentiostatic-based cycle. In the test, the charging voltage value was set to 0.5 V and 1.0 V, and the current density was respectively set to 1 Ag^−1^, 2 Ag^−1^, 5 Ag^−1^ and 10 A g^−1^. According to the GCD curve of S−iSkin under different current densities (see the Appendix A–h for the GCD curve), the analysis found that it has a lower voltage drop value under different current densities (see Appendix A). The GCD curve is processed by Equations (4)–(6) to obtain the specific capacitance, energy density and power density change curves under different current densities (see Figure 4g–i). It can be found that as the current density increases, the internal resistance of S−iSkin increased, and its specific capacitance and energy density showed a downward trend, while its power density showed an upward trend. Comparing the specific capacitance and energy density of S−iSkin with the others under the same current density, it can be found that it had a relatively large capacitance and energy density. This verified that it has certain advantages in processing and performance as well as the power output. According to the comparison of the power density of S−iSkin with the others under the same current density, it showed that S−iSkin had a higher power density, but it still had a certain advantage over some reported flexible sensors. Combined with the analysis of the CV and the EIS, it can be concluded that the doping of SA, NaCl and PANI significantly improved the electrochemical performance of S−iSkin, which led to a better electromechanical performance and sensing performance.

### 3.4. The Sensing Performance and Application of S−iSkin

In the experiment, S−iSkin (length × width: 70 mm × 15 mm) was applied with standard weights of different masses as the external stress. The detection of its sensing performance is completed by detecting the change value of the voltage after loading the stress. The study found that S−iSkin can stimulate a relatively weak stress (9.8 mN) and produce a significant voltage change (0.7 mV). As the magnitude of the loading stress gradually increases to 490 mN, the resulting voltage change also gradually increases to 54.1 mV (see Figure 5a). The sensitivity of S−iSkin under different loading stresses was analyzed by Equation (7), and it can be found that as the loading stress increases to 490 mN, the voltage change rate increases to 2.6 (see Figure 5b). By summarizing the rate of the voltage change generated by S−iSkin under different stresses, it can be found that it has good linearity under weak stress (see Figure 5c). Meanwhile, S−iSkin has excellent sensitivity (see Appendix A). In addition, S−iSkin can work stably for 2000 s under a stress of 198 mN, and can carry out more than 200 stress–voltage response cycles during this period (see Figure 5d) and the resistance of S−iSkin is relatively stable (see Appendix A). The excellent sensitivity, linearity, long service life and responsiveness of S−iSkin to micro-stress make it suitable for the preparation of skin sensing (see Appendix A) [31,32,33,34,35].

Since S−iSkin also has good biocompatibility and adhesion, it can also be attached to the force-receiving parts of the human body to monitor the movement of the human body. When the monitored person walked at a speed of 1.2 m/s, S−iSkin attached to the foot can effectively reflect the number of their steps and their travel speed (see Figure 5e). Similarly, when the monitored person ran at a speed of 3 m/s, it can also accurately monitor their exercise status (see Figure 5f). Therefore, during the human body’s movement to the external processing chip, a set of real-time human body motion monitoring systems can be prepared by connecting the sensor signal generated via S−iSkin. The real-time monitoring of human health in sports can be safely and effectively solved in this way.

## 4. Conclusions

In summary, we have developed a highly sensitive, ultra-durable, eco-friendly flexible ion-driven sensing ionic skin. Due to the network skeleton of starch–SA, it has obtained good stretchability (60% elongation at break) without regenerated substances (from FTIR and XRD). Then, its hydrophilicity is greatly reduced (57.5° of water contact angle) by the assistance of PANI’s hydrophobicity. Relying on the good conductivity of NaCl particles and the effective improvement of the conductivity of S−iSkin by PANI, the sensing performance and electrochemical performance of S−iSkin have been significantly improved. This allows S−iSkin to produce a voltage change of 54.1 mV and a voltage change ratio of 2.6 under the action of a micro-stress of 490 mN. In addition, it also has good service life and stability. The test found that it can work stably for 2000 s under a stress of 198 mN. Meanwhile, it can perform more than 200 stress–voltage response cycles during the test.

Therefore, this work proposes a S−iSkin with excellent sensitivity, linearity, good biocompatibility and a long service life. Meanwhile, it demonstrates a sensing form that can directly convert external stress stimuli into electrical signals without an external power supply and improve its sensing performance via improving its conductivity. The excellent performance of S−iSkin allows it to be applied to human movement monitoring and biological sign sensing. It can not only be applied to the research and development of flexible and wearable motion monitoring devices, but also can develop micro-stress sensing equipment applied to human social life. This provides the impetus for the development of flexible sensors in the direction of green environmental protection and close to the actual social life of human beings.

## Figures and Tables

**Figure 1 polymers-14-01902-f001:**
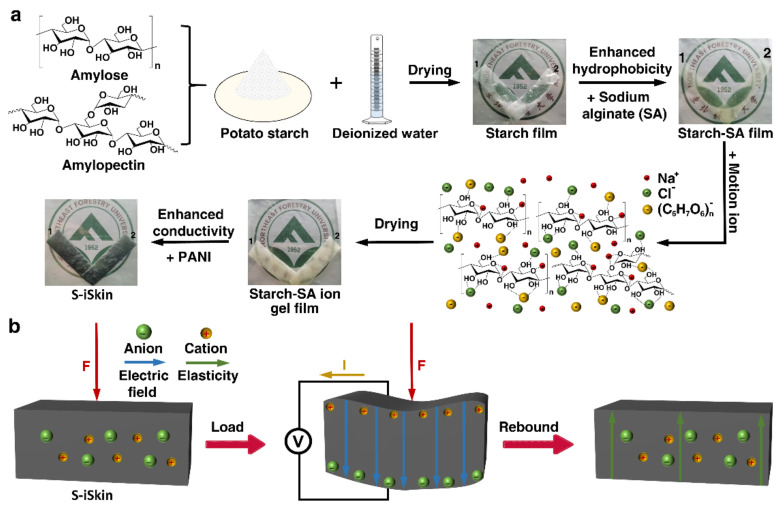
Schematic diagram of the synthetic preparation and working principle of S−iSkin: (**a**) Preparation process of S−iSkin. (**b**) Working principle of S−iSkin.

**Figure 2 polymers-14-01902-f002:**
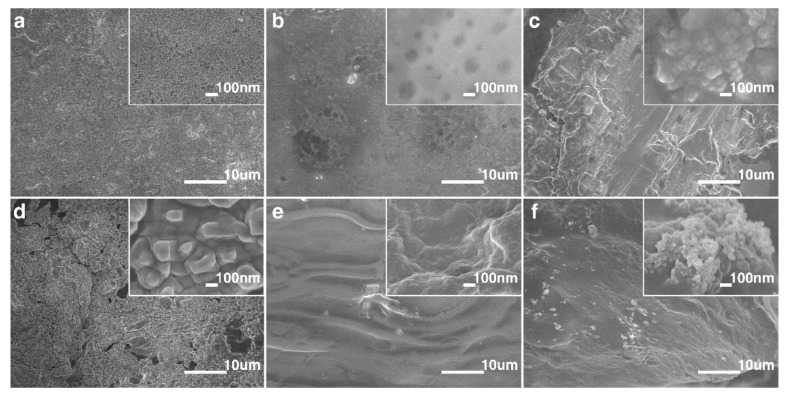
SEM image of S−iSkin: (**a**) the starch film. (**b**) the starch–SA film. (**c**) the ionic starch–SA film. (**d**) S−iSkin. (**e**) SEM image of cross section of the starch film. (**f**) SEM image of cross section of S−iSkin.

**Figure 3 polymers-14-01902-f003:**
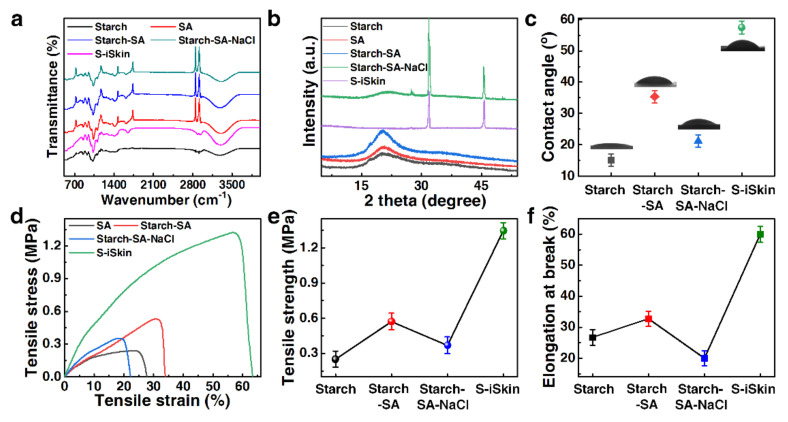
Experimental characterization of S−iSkin: (**a**) FT−IR curves, (**b**) XRD curves, (**c**) water contact angle, (**d**) the curve of stress−strain, (**e**) tensile strength, (**f**) elongation at break.

**Figure 4 polymers-14-01902-f004:**
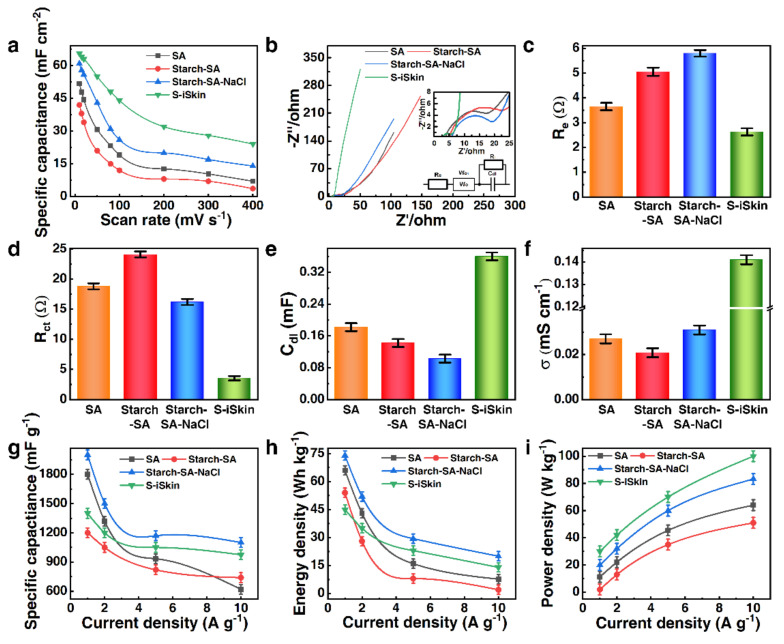
Electrochemical properties of S−iSkin: (**a**) Change curve of specific capacitance under CV test of S−iSkin. (**b**) The EIS curve of S−iSkin at 10^5^ Hz–10^−2^ Hz. (**c**) Equivalent resistance (R_e_) value of S−iSkin under EIS test. (**d**) Charge transfer resistance (R_ct_) value of S−iSkin under EIS test. (**e**) Capacitance (C_dl_) value of S−iSkin under EIS test. (**f**) The conductivity (σ) of S−iSkin under the EIS test. (**g**) The curve of specific capacitance and current density of S−iSkin under GCD test. (**h**) The energy density and current density curve of S−iSkin under GCD test. (**i**) The curve of power density and current density of S−iSkin under GCD test.

**Figure 5 polymers-14-01902-f005:**
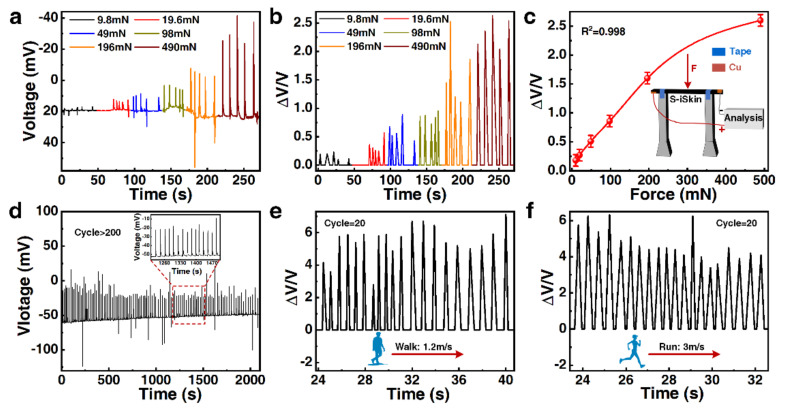
Sensing performance of S−iSkin and its application in human motion monitoring: (**a**) Changes in the response voltage of S−iSkin under different stresses. (**b**) The response voltage change rate of S−iSkin under different stresses. (**c**) The curve of the stress and response voltage change rate of S−iSkin. (**d**) The life test of S−iSkin. (**e**) The application of S−iSkin to human walking monitoring. (**f**) The application of S−iSkin to human running monitoring.

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
