# Peer review of "A Highly Sensitive, Ultra-Durable, Eco-Friendly Ionic Skin for Human Motion Monitoring"

_polymers, 2022, doi:10.3390/polym14091902_

Round 1
Reviewer 1 Report
Dear authors,
Thank you for submitting your manuscript entitled "A highly sensitive, ultra-durable, eco-friendly ionic" to MDPI. This paper developed an ionic skin (S-iSkin) based on edible starch-sodium alginate (Starch-SA), which can convert the external strain stimulus into a voltage signal without an external power supply.
The manuscript qualifies for publication following minor revisions.
Please address the following points:
1) Better state the novelty of your work at the introduction section.
2) Provide all material order numbers and manufacturers
Author Response
Reviewer #1
Thanks to the reviewer's suggestion. According to the reviewer's suggestion, we made the detailed correction to the article. The specific corrections are as follows:
(1) Better state the novelty of your work at the introduction section.
Response to (1): Thanks to your comments, we have made relevant revisions in the introduction section and highlighted the revised parts. The innovative nature of the S-iSkin is described in more detail.
Here, we propose a self-powered sensing ion-conducting hydrogel using highly environmentally friendly and excellent biomaterials, and based on its good adhesion and mechanical properties, it can be used for ion-skin preparation strategy. In this strategy, an ion-driven ion-skin (S-iSkin) based on edible starch-sodium alginate as a grid-like framework is proposed. At the same time, it has excellent sensitivity, linearity, long service life and good biocompatibility. These excellent properties allow S-iSkin to be effectively applied to detection possibilities such as human health monitoring. Compared with the existing electronic skins, it can be directly attached to the human skin surface through its own adhesion without external conditions and external power supply for motion monitoring, which makes the S-iSkin work without any damage to the human body. After failure, it can become ion-conducting hydrogel again by simple high temperature dissolution. These excellent properties indicate that our developed renewable biomass ionic skin has an important role in serving human health monitoring and micro-stress sensing devices.
(2) Provide all material order numbers and manufacturers
Response to (2): Thanks to your tip, we have corrected the information for all materials used in the experiments. And we have clearly marked the revised content.
The potato starch (Starch, CAS: 9005-25-8) and polyaniline (PANI, purity≥98%, CAS: 5612-44-2) were purchased from Cool Chemical Technology Company (Beijing, China) in the experiment. Sodium alginate (SA, purity≥99.9%, CAS: 9005-38-3) was ordered from Xiling Chemical Company (China, Guangdong), and sodium chloride (NaCl, purity≥99.9%, CAS: 7647-14-5) was acquired from Tianda Chemical Reagent Factory (China, Tianjin). Nano-silica (SiO2, particle size of about 30 nm, purity≥99.5%, CAS: 7631-86-9) was purchased from Keyan Industrial Co., Ltd. (Shanghai, China), and glycerol (purity≥99.9%, CAS: 56-81-5) was bought from Yongda Chemical Reagent Co., Ltd. (Tianjin, China). Deionized water (CAS: 7732-18-5) was obtained from Yongchang Reagent Company (Harbin, China). Cold field emission scanning electron microscope (SEM, SN-7500F, JEOL, Japan).
Reviewer 2 Report
In this work Zhuangzhi Sun et al., developed a highly and eco-friendly ionic conductive hydrogel that integrates sensing, durability, ion conductivity, and biocompatible performance, which could be serviced as ionic skin. The author claimed high sensitivity, linearity, long service life and good biocompatibility. Overall, the work seems interesting, however in the reviewer point of view some important discussions and investigations are missing, therefore some revisions should be important.
Comments
The rational or objective of the paper is not presented in a clear way. The authors should focus on that.
The authors should investigate the temperature stability of the ionic skin also. As for practical applications it is needed.
The response/recovery times should be highlighted and can be represented with a separate plot, as this is important parameter of any type of sensor.
The authors should address the stability of the ionic skin in detail also.
The author claimed that “these findings provide a foundation for the development of health monitoring systems” therefore the applications of these devices in sensing radial artery pulses, acoustic vibrations, and human body motion can be investigated.
The authors should put all the reportable data in tabulated form, and should compare this with relevant reports.
Author Response
Reviewer #2
Thanks to the reviewer's suggestion. According to the reviewer's suggestion, we made the detailed correction to the article. The specific corrections are as follows:
(1) The rational or objective of the paper is not presented in a clear way. The authors should focus.
Response to (1): Thanks to your suggestion, we have revised the relevant content. Supplementary clarifications have been made for unclear points in the manuscript. At the same time, we also supplement the relevant experiments in the supporting information to fully justify the paper. All modifications have been highlighted.
Here, we propose a self-powered sensing ion-conducting hydrogel using highly environmentally friendly and excellent biomaterials. Based on its good adhesion and mechanical properties, it can be used for ion-skin preparation strategy. In this strategy, an ion-driven ion-skin (S-iSkin) based on edible starch-sodium alginate as a grid-like framework is proposed. Meanwhile, it has excellent sensitivity, linearity, long service life and good biocompatibility. These excellent properties allow S-iSkin to be effectively applied to detection possibilities such as human health monitoring. Compared with the existing electronic skins, it can be directly attached to the human skin surface through its own adhesion without external conditions and external power supply for motion monitoring, which makes S-iSkin work without any damage to the human body. After failure, it can become ion-conducting hydrogel again by simple high temperature dissolution. These excellent properties indicate that our developed renewable biomass ionic skin has an important role in serving human health monitoring and micro-stress sensing devices.
(2) The authors should investigate the temperature stability of the ionic skin also. As for practical.
Response to (2): Thanks to your suggestion, to verify the temperature stability of our proposed ionic skin, we have added the relevant thermal analysis experiments (see Figure 1R). The thermogravimetric analysis curve shows that the decomposition temperature of ionic skin is about 170 °C, which is sufficient to meet the needs of its normal working environment temperature on the surface of human skin. In the initial stage, its quality showed a downward trend, mainly due to the evaporation of its internal water. Since our ionic skin is highly hydrophilic, it can absorb sweat from human skin, so its internal moisture evaporation has little effect on its performance.
Figure. 1R. The thermogravimetric analysis of the S-iSkin.
(3) The response/recovery times should be highlighted and can be represented with a separate plot, as this is important parameter of any type of sensor.
Response to (3): Thanks to you for your comments, we have added the relevant content to the supporting information section of the article (see Figure 2R). The slowest response time of S-iSkin is 0.92s (under a stress of 98mN) and the response time of S-iSkin does not exceed 1s under other stresses. The slowest recovery time is 2.42s (under a stress of 490mN), and the recovery time increases with the increase of stress. This is mainly because the deformation amount of S-iSkin increases with the increase of stress, so it takes longer time to recover to the initial state under the action of large stress. But in general, its recovery time can be guaranteed to be less than 1s within a stress of 49 mN.
Figure. 2R. (a)-(b) Response time and recovery time of the S-iSkin.
(4) The authors should address the stability of the ionic skin in detail also.
Response to (4): Thanks for your suggestion, we have supplemented the manuscript with a note on stability. First, the S-iSkin has a long service life, the S-iSkin can work stably for 2000 s under a stress of 198 mN, and can carry out more than 200 stress-voltage response cycles during this period (see Figure 5(d)). Secondly, in the use test experiment of the S-iSkin, it can be found that its voltage response is more obvious and the variation range is more stable. In addition, it can be seen from its thermal stability test that it begins to decompose at 170 °C, so it can maintain a stable state during human motion monitoring. At the same time, the S-iSkin has a small resistance change during water loss (see Figure 3R). Under the light intensity of 1.2 kW m-2, the resistance change rate per unit area in one hour is only 9.1%. To sum up, the S-iSkin has good stability.
Figure. 3R. (a) Resistance of S-iSkin. (b) Variation of resistance value of S-iSkin with evaporation of water. (c) The water loss rate of the S-iSkin. (d) The relationship between the water loss of the S-iSkin and its resistance value.
(5) The author claimed that “these findings provide a foundation for the development of health monitoring systems" therefore the applications of these devices in sensing radial artery pulses, acoustic vibrations, and human body motion can be investigated.
Response to (5): Thanks to your comments, we have supplemented the experiments of the application of S-iSkin in sensing radial artery pulse, the details of which are shown in Figure 4R. It can be found from Figure 4R (a)that the pulse of the subjects measured by S-iSkin is 71 times min-1, while the pulse of normal people is 70-80 times min-1, which is in line with human health parameters. The average voltage change rate is 0.47, and the change is relatively stable, which is convenient for data collection. Due to the limitations of the existing conditions in our laboratory, the detection of acoustic vibrations and other human motions is not yet possible.
Figure. 4R. (a)-(b) Pulse monitoring by S-iSkin
(6) The authors should put all the reportable data in tabulated form, and should compare this with relevant reports.
Response to (6): Thanks to your comments, we have reviewed the relevant literature to summarize the existing reports on ionic skin. Through correlation comparison, it can be found that our S-iSkin has obvious advantages in sensitivity (see Table 1R).

Reviewer 3 Report
The manuscript titled as ‘A highly sensitive, ultra-durable, eco-friendly ionic skin’ focuses on the preparation of ionically conductive ionic hydrogels that can be used as an ionic skin. Synthesis, characterization and application of the produced samples are provided.
Please see below the following comments and remarks associated with the manuscript:
- Title of the manuscript is incomplete. This has to be corrected.
- The manuscript needs a further language editing.
In section 3.2. structural characterization and mechanical properties
- The FTIR data provided by the authors have to be discussed more in detail. The signals have to be assigned to the associated vibrations and discuss the differences and probable causes. The assignments can be given in the supporting section.
- What are the probable causes for better mechanical performance of the S-iSkin? Some discussion needs to be inserted to improve the structure property understanding.
- What is the time dependent conductivity of the S-iSkin? How does it change with time as the water evaporates from the system? Or vice versa.
Kind regards.
Author Response
Reviewer #3
Thanks to the reviewer's suggestion. According to the reviewer's suggestion, we made the detailed correction to the article. The specific corrections are as follows:
(1) Title of the manuscript is incomplete. This has to be corrected.
Response to (1): Thanks to the comments of the reviewer, we have revised the title to " A highly sensitive, ultra-durable, eco-friendly ionic skin for human motion monitoring".
(2) The manuscript needs a further language editing.
Response to (2): Thanks so much for your constructive suggestion. We have re-edited language and grammar in full article.
(3) In section 3.2. structural characterization and mechanical properties The FTIR data provided by the authors have to be discussed more in detail. The signals have to be assigned to the associated vibrations and discuss the differences and probable causes. The assignments can be given in the supporting section.
Response to (3): Thanks to your reminder, we have supplemented the analysis section of the FT-IR data in the manuscript. At the same time, we have discussed in detail the differences between different samples and the possibility of causing the differences.
Compared with the Starch-SA-NaCl film in Fig. 3(a), the disappearing peaks of S-iSkin at 2913cm-1, 2848cm-1, 1734cm-1 and 1472cm-1 are the vC-H asymmetric stretching vibration peak of the unsaturated group, the vC-H symmetric stretching vibration peak of the unsaturated group, the vC=O stretching vibration peak of the carboxyl group and the vC-C unsaturated ring stretching vibration peak. This is mainly due to the strong oxidizing property of PANI, which can oxidize the unsaturated functional groups in SA, so that the peaks at the above positions disappear. However, the S-iSkin has a vN-H stretching vibra-tion peak at 1644 cm-1, which proves that PANI can exist relatively stably in S-iSkin. In addition, since the unsaturated functional groups of SA are largely oxidized, the performance of S-iSkin is closer to that of starch films, and its chemical properties are more stable and have a better degree of degradation.
(4) What are the probable causes for better mechanical performance of the S-iSkin? Some discussion needs to be inserted to improve the structure property understanding.
Response to (4): Thanks to your comments, the reason for the enhanced mechanical properties of the S-iSkin have been supplemented in the article. This is mainly because the S-iSkin has a grid-like skeleton, which makes it have a large specific surface area. In addition, the introduction of PANI effectively reduced the crystallization of large NaCl particles, enabling them to be uniformly dispersed into Starch-SA, reducing the formation of NaCl crystal nuclei. At the same time, as a high-molecular polymer, the highly conductive PANI can also have a strong van der Waals force with the high-molecular polymer (Starch and SA) [29, 30], which is higher than the van der Waals force in the Starch-SA film. Therefore, under the combined action of three factors, the mechanical properties of S-iSkin have been greatly improved. This part of the revision has been highlighted in the article.
(5) What is the time dependent conductivity of the S-iSkin? How does it change with time as the water evaporates from the system? Or vice versa.
Response to (5): Thanks to you for your comment, we have added the relevant experiments in the supporting information. Figure 3R (a) shows that the resistance of the S-iSkin is relatively stable, and its resistance value per unit area is about 0.26MΩ, which is similar to the conductivity value of the S-iSkin obtained by our EIS test. In addition, in view of the fact that the S-iSkin will lose water during actual use, we use the CEL-SA500/350 xenon lamp to simulate the evaporation environment to test the water loss of the S-iSkin under the light intensity of 1.2 kW m-2 condition and changes in its resistance value. It can be found that the resistance value of the S-iSkin decreases with the evaporation of water (see Figure 3R (b)), mainly because it does not remain in the state of hydrogel as the water evaporates. Furthermore, the water molecules inside the S-iSkin are gradually removed, and the existence of these water molecules hinders the electron transfer inside the S-iSkin. Therefore, the resistance value of the S-iSkin decreases with the evaporation of water.
From Figure 3R (c), it can be seen that the water loss rate of the S-iSkin decreases with time increasing. By summarizing the relationship between the water loss of the S-iSkin and its resistance value (see Figure 3R (d)), it can be found that with the increase of the water loss of the S-iSkin, its resistance value gradually decreases. When it is transformed from a hydrogel state to a solid thin film state, its resistance value drops sharply, but its resistance value changes less. Its resistance per unit area decreased by only 9.1% in 1h. In addition, the S-iSkin has excellent hydrophilicity, it can also absorb the moisture in human skin and the air during the working process, so it can stably monitor human movement.
Figure. 3R. (a) Resistance of S-iSkin. (b) Variation of resistance value of S-iSkin with evaporation of water. (c) The water loss rate of the S-iSkin. (d) The relationship between the water loss of the S-iSkin and its resistance value.

Round 2
Reviewer 2 Report
The revised manuscript is reflecting the feedback.